# Real-World Evidence on Disease Burden and Economic Impact of Sickle Cell Disease in Italy

**DOI:** 10.3390/jcm12010117

**Published:** 2022-12-23

**Authors:** Lucia De Franceschi, Chiara Castiglioni, Claudia Condorelli, Diletta Valsecchi, Eleonora Premoli, Carina Fiocchi, Valentina Perrone, Luca Degli Esposti, Gian Luca Forni

**Affiliations:** 1Department of Medicine, University of Verona, 37129 Verona, Italy; 2Novartis Farma S.p.A., 20154 Milan, Italy; 3CliCon S.r.l. Società Benefit Health, Economics & Outcomes Research, 40137 Bologna, Italy; 4Centro della Microcitemia e Anemie Congenite, Ospedale Galliera, 16128 Genoa, Italy

**Keywords:** administrative databases, real-life, SCD epidemiology, vaso-occlusive crisis, transfusions

## Abstract

A real-world analysis was conducted in Italy among sickle cell disease (SCD) patients to evaluate the epidemiology of SCD, describe patients’ characteristics and the therapeutic and economic burden. A retrospective analysis of administrative databases of various Italian entities was carried out. All patients with ≥1 hospitalization with SCD diagnosis were included from 01/2010-12/2017 (up to 12/2018 for epidemiologic analysis). The index date corresponded to the first SCD diagnosis. In 2018, SCD incidence rate was 0.93/100,000, the prevalence was estimated at 13.1/100,000. Overall, 1816 patients were included. During the 1st year of follow-up, 50.7% of patients had one all-cause hospitalization, 27.8% had 2, 10.4% had 3, and 11.1% had ≥4. Over follow-up, 6.1–7.2% of patients were treated with SCD-specific, 58.4–69.4% with SCD-related, 60.7–71.3% with SCD-complications-related drugs. Mean annual number per patient of overall treatments was 14.9 ± 13.9, hospitalizations 1.1 ± 1.1, and out-patient services 5.3 ± 7.6. The total mean direct cost per patient was EUR 7918/year (EUR 2201 drugs, EUR 3320 hospitalizations, and EUR 2397 out-patient services). The results from this real-world analysis showed a high disease burden for SCD patients with multiple hospitalizations during the follow-up. High healthcare resource utilization and costs were associated with patient’ management and were most likely underestimated since indirect costs and Emergency Room admissions were not included.

## 1. Introduction

Sickle Cell Disease (SCD) is a worldwide distributed hereditary red cell disorder recognized by WHO as an emerging global health problem. SCD is caused by a point mutation occurring in the β-globin gene, resulting in the synthesis of pathological hemoglobin S (HbS) [1]. This protein polymerizes upon deoxygenation, causing red blood cell sickling and dehydration [2,3]. The vaso-occlusion process and the pathophysiological aspects of the disease is the result of more complex interactions between the pro-adhesive sickle cells, neutrophils, and endothelium that trigger a pro-inflammatory environment [3,4,5,6]. The main clinical manifestations of SCD are chronic hemolytic anemia and acute vaso-occlusive crisis (VOCs), which are associated with severe pain and amplified inflammatory response, resulting in end organ damage [7]. This still impacts morbidity and mortality of patients with SCD [8], as well as patients’ quality of life [7].

The therapeutic management of SCD span in the following main areas [9]: supportive care as newborn screening; the prevention of infection (penicillin prophylaxis, vaccination); the management of acute and chronic complications; and disease-modifying therapies specific for SCD, such as hydroxyurea and more recently Crizanlizumab, Voxelotor and L-Glutamine [9,10,11]. Although the therapeutic portfolio of SCD is increasing, the benefits of novel therapeutic options on disease natural history still needs to be established. This highlights the importance of real-life studies on SCD clinical management and outcomes. In addition, the generation of knowledge on disease burden in different care settings might help (i) to prevent undertreatment of adult SCD patients with increased risk of severe organ dysfunction and (ii) to reduce costs linked to SCD patients’ frequent emergency department visits [12,13,14].

In Europe, SCD is now regarded as the paradigm of immigration hematology [15]. Indeed, SCD was originally limited to sub-Saharan African areas; however, migration fluxes have significantly impacted SCD distribution and increased its prevalence in areas where it was previously uncommon [16]. In this scenario, recent immigration from Africa, South America, and the Balkans has changed the geographic profile of SCD, thus also becoming an emerging public health burden in Italy, enlarging the endemic SCD population and with an estimate of 3–5-fold increased prevalence over the last years [9].

A national register of hemoglobinopathies was approved in 2017 in Italy, but it has not yet been fully populated with clinical data of all patients with hemoglobinopathies cared for by the national territory [17]. Therefore, we designed a retrospective observational study, the GREATALYS (Generating Real world Evidence Across Italy In SCD, study code CSEG101AIT01), on Italian health claims data to estimate the epidemiologic burden and the economic impact of SCD in Italy. The main purpose was to evaluate the epidemiology of SCD in Italy in terms of prevalence and incidence rate and to provide an estimation on the number of patients currently living with SCD. The secondary aims were to describe the demographic characteristics of patients affected by SCD and to analyze the patients’ journey in terms of therapeutic pathways, disease features, and economic insights in clinical practice setting by using real-world data.

## 2. Materials and Methods

### 2.1. Data Source

In Italy, healthcare is provided by a public system, known as the National Health System (NHS), which is based on the principle of universal coverage for all citizens and residents, namely health-assisted subjects. This study is based on existing data linked and extracted from administrative databases used by the Italian NHS for the reimbursement of healthcare services of two Regional and fifteen Local Health Units geographically distributed throughout the Italian territory, covering approximately 15.3 million health-assisted subjects (about 25% of the whole Italian population). According to the objectives of the analysis, the following databases and related items were used: beneficiaries’ database, which contains all demographic data for patients in analysis; pharmaceutical databases that collect data on all drugs supplied through Anatomical Therapeutic Chemical code (ATC), marketing authorization holder code, the number of packages and number of units per package, unit cost per package, and prescription date; hospitalization database that includes all hospitalization data with discharge diagnosis codes classified according to the International Classification of Diseases, Ninth Revision, Clinical Modification (ICD-9-CM), diagnosis Related Group (DRG) and DRG-related charge (provided by Health System); outpatient specialist services database, which contains date of prescription, type, and description activity of diagnostic tests and visits for patients in analysis and laboratory test or specialist visit charge. Emergency Room, “Temporarily Present Foreigner” codes, and refugees were not recorded in this study. In order to guarantee patient privacy, each beneficiary was assigned an anonymous univocal numeric code (patient ID) by the Region/Local Health Units that allowed electronic linkage between databases. The anonymous code of the patient ensures the anonymity of the extracted data in full compliance with EU Data Privacy Regulation 2016/679 (“GDPR”) and Italian D.lgs. n. 196/2003, as amended by D.lgs. n. 101/2018. All the results of the analyses were produced as aggregated summaries, which could not be connected, either directly or indirectly, to individual patients. Informed consent was not required since obtaining it would be impossible for organizational reasons (pronouncement of the Data Privacy Guarantor Authority, General Authorization for personal data treatment for scientific research purposes—n. 9/2014). The study was approved by the Ethics Committee of the Regions and Local Health Units involved.

### 2.2. Study Design

All patients with main or secondary discharge diagnosis for SCD were included between 1 January 2010 to 31 December 2017 (up to December 2018 for epidemiologic analysis). SCD diagnosis was extrapolated from the hospitalization database through ICD-9-CM codes (provided in Appendix A) previously reported in the literature [18,19,20]. Regarding the epidemiologic analysis, prevalence, and incidence rates were stratified by age, sex, and the presence/absence of VOCs at inclusion (defined by ICD-9-CM code for SCD with crisis, as explained in the next section). Incidence was defined as the number of new SCD hospitalization discharge diagnoses with codes listed above within each year of inclusion (considering all data availability period). Data were also re-proportioned to fit the overall Italian population for the year 2018: the projections were obtained by reproportioning the epidemiological data of the sample population in the analysis to the national scale, as previously described [21,22]. For longitudinal analyses, the index date corresponded to the first diagnosis detected in the database between 1 January 2010 to 31 December 2017 (inclusion period). Patients were followed up from the index date to death or end of data availability (at least one year). A one-year period before the index date was used to characterize patients. Patients without data available in the one-year period prior the index date or who moved away from the Region/Local Health Units after the index date were excluded from the analysis.

### 2.3. Study Variables

At index date, all included patients were described according to demographic characteristics, such as age and gender, expressed by proportion of female subjects. Clinical characteristics at baseline were recoded using the Charlson comorbidity index, which assigns a score to each concomitant disease assessed in the previous 12 months on drug treatment and hospitalizations; therefore, patients who were untreated and not hospitalized with comorbidities were not captured [23]. The patient’s distribution by type of SCD diagnosis at index date was also reported. During follow-up, therapeutic pathways were evaluated as all treatments prescribed (by third level ATC code) and all SCD treatments classified as follows: SCD-specific treatment, SCD-related drugs, and SCD complication-related drugs; detail on the type of drugs within each category is provided in Appendix A. Blood transfusions were identified by presence of ICD-9-CM code V58.2 (blood transfusion, without reported diagnosis) or ICD-9-CM procedure code 99.0 (transfusion of blood). To obtain insights into patients’ journey, all-cause hospitalizations were grouped by Major Diagnostic Category (MDC) and SCD-related hospitalization were analyzed (both ordinary admissions and day hospitals were considered; Emergency Room access not requiring hospitalizations were not included). VOCs and SCD complications were also considered by applying the same methodology reported in the literature [7]. These were identified by ICD-9-CM codes for SCD with crisis and VOC claims without a three-day gap were combined and considered as one single VOC episode. Therefore, only VOC requiring hospitalization were captured in this study. SCD complication codes are reported in Appendix A.

Mean annual healthcare resource consumption and related direct costs sustained by the NHS were analyzed during follow-up in terms of overall treatments, all-cause hospitalizations, and out-patients services. The costs are reported in Euros (EUR). Drug costs were evaluated using the Italian NHS purchase price. Hospitalization costs were determined using the DRG tariffs, which represent the reimbursement levels of the NHS to healthcare providers. The cost of instrumental and laboratory tests was defined according to the regional tariffs. The transfusion costs refer to costs for the procedure and did not include all other costs regarding the overall transfusion processes, for instance those related to transformation activity (blood collection, validation, and donor services).

### 2.4. Statistical Analysis

Descriptive analyses are presented in this report. Categorical data were summarized in terms of the number and percentages of patients providing data. The frequency counts and percentages of patients in each category are provided. Percentages were calculated using the number of observations with non-missing values as the denominator. Continuous data were summarized in terms of mean, with standard deviation (SD). All analysis was performed using STATA SE version 12.0.

## 3. Results

### 3.1. Epidemiology Estimates a High Prevalence of SCD in Italy

An epidemiological analysis was first carried out to evaluate the incidence and prevalence of SCD diagnosis within the sample population. Incidence rates slightly fluctuated over the years, ranging from 2.89 per 100,000 health-assisted subjects in 2010 to 0.93 per 100,000 in 2018. Since the data were available starting from 2009, the higher incidence observed for the year 2010 could be due to prevalent patients for whom only one year was available to distinguish between prevalent and incident patients. In the last year of inclusion, the incidence rate of SCD was 1.09 per 100,000 among females and 0.76 per 100,000 among males (Figure 1A). The prevalence of SCD for 2018 was of 13.1 cases per 100,000 health-assisted subjects, 10.9 cases per 100,000 males, and 15.3 per 100,000 females. SCD prevalence in younger (<18 years old) and adult (≥18 years old) individuals was 17.2 and 12.4 cases per 100,000, respectively (Figure 1B). The prevalence of adult SCD patients without VOCs at diagnosis in 2018 ranged from 6.99 (age group 25–29) to 11.63 (age group 75–84) per 100,000 health-assisted subjects without a clear pattern, while the prevalence of adult SCD patients with VOCs at diagnosis tended to be higher among young adults (2.81 and 3.04 in age ranges 18–24 and 25–29, respectively to a prevalence below 1 for age group from 55 to 84 years old) (Figure 1C). According to geographic distribution, the prevalence of SCD with crisis was 2.96 in the north, 1.77 in the central, and 2.06 in the south regions of Italy per 100,000 health-assisted subjects (Figure 1D).

The number of SCD patients were then projected on the Italian population of year 2018, estimated as 59,816,673 inhabitants, according to the yearly ISTAT (Italian National Institute of Statistics) report (available online: https://www.tuttitalia.it/statistiche/popolazione-andamento-demografico/ (accessed on 20 October 2022). A total of 7977 patients with SCD was estimated for year 2018, specifically 1690 young and 6287 adult patients. Stratification by gender estimated a total of 3235 male (919 young and 2316 adult) and 4742 female (771 young and 3971 adult) SCD patients among the Italian population (Figure 1E). Among adult patients with SCD, the age-group more represented were 35–44 years (N = 1273) and 45–54 years (N = 1149) (Figure 1F). Distribution by type of diagnosis revealed 1279 patients out of 7977 are estimated to have a diagnosis of SCD with concomitant VOC and 5894 to have SCD without concomitant VOC (Figure 1G). Taken together, these data suggest an increasing prevalence and incidence of SCD in Italy and underline a sub-population of SCD patients with a milder clinical phenotype not being referred to comprehensive centers for hemoglobinopathies.

### 3.2. High Healthcare Utilization and Significant Direct Medical Costs Contribute to High Burden of Sickle Cell Disease in Italian Adult Population

A total of 2347 patients matched at least one inclusion criterium during all study periods (Figure 2). Of these, 1816 (77.4%) presented the 12-month period minimum before the index date needed for characterization and a 12-month period minimum after the index date to allow follow-up analysis and were therefore included.

Table 1 details the demographic and clinical characteristics of SCD patients included in the analysis. The average age was 43.8 years and 58.4% were female. Adults with SCD had a Charlson comorbidity index value that was higher than younger patients, defined as subjects under 18 years of age (1.2 and 0.4, respectively). Regarding the diagnosis at index date, 74.3% were SCD without concomitant VOCs, 16.1% with concomitant VOCs, and 9.6% were unspecified.

The distribution of SCD genotypes among patients was 70.9% Sβ genotype without crisis; 9.7% unspecified SCD genotype with crisis; 6.9% of Sβ genotype with crisis; 4.6% and 2.0% Hb-SS with and without crisis, respectively; 3.9% other SCD with crisis (0.6% other SCD without crisis); and 0.8% and 0.5% SCD genotype without and with crisis, respectively. In patients without crisis, SCD was mainly included through secondary diagnosis (83.7% of patients without crisis), while patients with crisis were mainly identified by primary diagnosis (59.4% of patients with crisis). In SCD patients without crisis and included by secondary SCD diagnosis, a wide heterogeneity of primary diagnoses was collected. Considering the most frequent ones, 2.1% of SCD patients had a primary diagnosis for obstructive chronic bronchitis with (acute) exacerbation, 1.7% for iron deficiency anemias secondary to blood loss (chronic), while atrial fibrillation or unspecified chest pain accounted for 1.4% and 1.2% of primary diagnoses (Figure 3A). To address the question as to whether knowledge on SCD might affect the use of ICD-9 codes for SCD at hospital discharge, we looked for departments where the most frequent admission/discharge hospital areas were reported. Hematology and general medicine were found to be the main hospital areas involved (evaluated in a sub-cohort of 927 patients for whom these data were available). Physicians from both areas are the specialists commonly more experienced in SCD management with its clinical manifestations. During first year of follow-up (index date included), 50.7% of patients had one all-cause hospitalization, 27.8% had two, 10.4% had three and 11.1% had ≥four. In the second year of follow-up, 44% had at least one hospitalization, and in the third and fourth year that number dropped to 38.2% and 35.8%, respectively (Figure 3B).

The length of hospitalization stay was of 8.0 ± 7.2 days for ordinary hospitalizations, with those related to musculoskeletal system having a mean length from a minimum of 9.1 days and up to 20.1 days. VOCs and SCD-related complications were evaluated during follow-up (SCD index hospitalization was excluded). Since the Emergency Room database was not among the databases in analysis, there might be an underestimation of acute events requiring intensive medical treatment (e.g., infusion for the hydration of pain treatment) [24]. Here, we focused on the hospitalization of patients with sickle cell-related acute events and SCD complications, reported as discharge diagnosis. Considering all available follow-up (mean 4.9 ± 2.2 years), the proportion of those with one VOC/SCD complication was 3.7% and 15.2%, with two VOC/SCD complications was 2.3% and 6.2%, and with ≥3 VOC/SCD complications was 7.7% and 12.8%, respectively (Figure 3C).

As expected, the most frequent ordinary hospitalization regarded the Major Diagnostic Category (MDC) blood and blood-forming organs and immunological disorders, which include SCD diagnosis, followed by those related to circulatory system (range between 12% during first year to 2.6% during fourth year of follow-up) and respiratory system (range between 10.8% during the first year to 2.4% during third year of follow-up) (Table 2).

SCD related treatment patterns are reported in Figure 4A,B. The proportion of patients with SCD-specific treatment, i.e., with hydroxyurea, a gold standard treatment for SCD, ranged between 6.1% and 7.2% during follow-up (Figure 4A). SCD-related drugs were prescribed to 58.4–69.4% of patients and those SCD-related complications rose from 60.7% to 71.3% (Figure 4A). Antithrombotic agents were prescribed in around 24.5–33.6% of patients each year of follow-up. Focusing on patients with VOC at index date, SCD disease-modifying drugs prescribed during follow-up were in the range 17.2–21.8%, SCD-related drugs were in the range 39.4–53.6% while SCD complication-related drugs were around 43.3–58% (Figure 4B). Mean total all-cause health care costs per patient during all available follow-up was EUR 7918, of which EUR 2201 was for overall drugs, EUR 3320 was for all-cause hospitalizations, and EUR 2397 was for outpatient specialist services (Figure 4C). It is noteworthy that around 15 treatment prescriptions per year were reported, a mean number of 1.1 hospitalizations and 5.3 outpatient services (Figure 4C).

During the 4 years of follow-up, among SCD-related drugs, anti-bacterial molecules were the most frequently prescribed (62.8–52.8%), followed by anti-inflammatory drugs (23.5–28.9%), and lastly analgesics (9–11%) with opioids prescribed only in around 8.0–10.5% of patients (Table 3).

Considering all available follow-up, the average yearly number of transfusions was 0.8 in the total population included, 1.4 in patients with crisis, 0.5 in young patients (<18 years), and 0.9 in adults. The mean annual cost for transfusions during all available follow-up was EUR 1291.

## 4. Discussion

Our study on real-world-data shows that the prevalence of SCD in Italy is higher than has been previously described [9]. This agrees with a previous report on prevalence of SCD in France [25], which examined the database that explores hospital access for sickle cell-related acute events [25]. Based on hospitalization discharge diagnosis, we found an incidence rate of 0.93 SCD cases per 100,000 and a prevalence of 13 SCD cases per 100,000 health-assisted individuals for year 2018, corresponding to 567 new SCD patients for year 2018. This highlights an increasing trend, which is similar to that reported by Hansen, D.L., et al. in the Danish population between 2000–2015 [26]. In addition, growing evidence generated by isolated studies on newborn screening in different Italian areas supports the increase trend of patients with SCD [27,28,29]. A world-wide estimation of inherited red cell disorders using global health data exchange database shows a significant increase in the global incidence of SCD associated with decreased mortality [30]. This characterizes Africa, which is continent more largely affected by voluntary migration or displacement towards Europe and North America due to violent conflicts. Our analysis provides an interesting focus on Italy, the gateway to Europe, reflecting the new world migratory influences and anticipating the impact of SCD on health systems. To better understand the phenomenon, the number of SCD patients were projected on the Italian population: a total of 7977 patients was estimated for year 2018 (1690 young and 6287 adult patients) [9]. Prevalence by geographic areas was in agreement with the presence of SCD patients in urban areas (northern regions) and historic SCD patients (southern regions) [15,31]. Approximately two thirds (74%) of the patients had at least one hospitalization without a concomitant diagnosis of crisis, thus suggestive of a high disease burden beyond severe VOC. In this direction, the International Sickle Cell World Assessment Survey reported a high prevalence of other symptoms, such as notable fatigue or depression/anxiety, that are perceived as high severe as VOCs by SCD patients [32].

Since the study population was identified by diagnosis codes of SCD, patients with milder forms not yet referred to specialist centers might have been included. Therefore, the sample analyzed might represent the pathology as a whole in terms of clinical variability and disease presentation. This might also explain the different estimates between our findings and the epidemiology reported: given the clinical variability of SCD, characterized by severe acute events as well as silent, progressive, and chronic course, real-world settings might capture those patients not taken over by specialist centers and therefore underreported in epidemiology studies.

The main cause of hospitalization agrees with other reports in similar European setting, confirming anemia, cardiopulmonary disease, and infections as the top three causes of hospitalization of patients with SCD [25]. Indeed, antibacterial drugs were among the most frequent drugs administered during characterization and follow-up periods, in line with the literature reporting that infections are a main complication of young adult and adults with SCD (from low urinary infection, tooth infection, pneumonia) [33,34,35,36,37,38]. This is also supported by the observation that the average number of 15.5 prescriptions/year is suggestive for a complicated and chronic clinical course. The relative low prescription of opioids might be most likely related to the lower use of major opioids alone according to the Italian guidelines on the management of pain [39,40,41,42]. This is based on multimodal analgesic therapy, which consists of the simultaneous use of molecules with different mechanisms of action, targeting pain of different origins. Multimodal analgesic therapy allows an optimal pain control, reducing the pharmacologic adverse events and attempting to avoid the development of chronic pain.

It is noteworthy that the use of drugs and the frequency of use decreases as we move away from the index hospitalization but the percentages of use generally remain high, especially with regard to anti-bacterials, anti-acids, anti-thrombotics, and anti-inflammatory. Concerning the gold-standard treatment for SCD, such as hydroxyurea or transfusion, we found a time-dependent reduction in hydroxyurea prescription in our population after hospital discharge. This might be related to the low compliance of the SCD population to hydroxyurea as chronic treatment as well as possible social barriers affecting the accessibility of migrants to public health services, such as outpatient clinics or family doctors [43,44]. In our study, the mean annual number of transfusions tend also to be higher among patients with crisis; in this regard, a recent Italian study pinpoints transfusion regimens to still be crucial as intensive treatments for both acute and chronic SCD-related complications [45].

Our study has some limitations, such as (i) the lack of detailed clinical information (e.g., genotypes, comorbidity, lifestyle habits) [46]; (ii) the full traceability of drugs (e.g.,: dosage prescribed, the actual use by the patients, and the therapeutic indication); and (iii) the exclusion of the Emergency Room database that might contribute to underestimating the phenomenon, as well as immigrants not coved by public health system.

Our estimates in the number of SCD patients could have considered patients not referred to specialist centers who are generally underreported, providing up-to-date insights into SCD profile and contributing to the determining of the extent, distribution, and burden of SCD in Italy. In this way, our data could be representative of the different presentation of the disease showing a high disease burden beyond VOCs that suggests a severe chronic clinical course of SCD, as also supported by the proportion of patients with anti-thrombotic agents as indicators of the underlying inflammatory vasculopathy and thrombotic risk that shed light on the management of patients not generally described in the literature. Last but not least, our data describe a population of SCD patients with high resource absorption, both in terms of drugs (for SCD and related diseases) but also for hospitalizations and specialist services. In comparison with other disorders the economic burden of which has been well investigated, such as myocardial infarction or cystic fibrosis, characterized by annual direct total healthcare costs/patients averaging EUR 3044–3668 [47] or ranging EUR 4000–30,000 (based on cystic fibrosis severity) [48], respectively, the cost of care for SCD remains constrained, with approximately EUR 8000 per year per patients. Moreover, the economic burden of SCD patients reported in the present analysis could be underestimated, since the healthcare resource consumption related to the access to emergency units in acute phase patients was not retrievable from the database in question. Ultimately, during the period considered, the only SCD-specific drug available was hydroxyurea, characterized by its low cost. The introduction of innovative therapies with different costs could modify the economic scenario of the overall direct healthcare expenditure for SCD, although it should be noted the increase in the drug costs could potentially correspond to a reduction in disease management costs.

## 5. Conclusions

Although population projections might present some limitations related to the time frame analyzed, the “starting” population for the projection, and the assumption about future trends, population projections represent a powerful tool for the development of strategies to intersect missing patients with SCD and define plans and policies for care of patients with SCD. This might help to ensure the timely management of patients with milder forms to prevent and treat clinical manifestations that could lead to more severe complications and/or more severe and potentially fatal clinical forms [49]. Finally, our data further reinforce the importance of a holistic approach to patients with SCD [32], who display a severe disease burden, which deeply affects their quality of life and require personalized illness education.

## Figures and Tables

**Figure 1 jcm-12-00117-f001:**
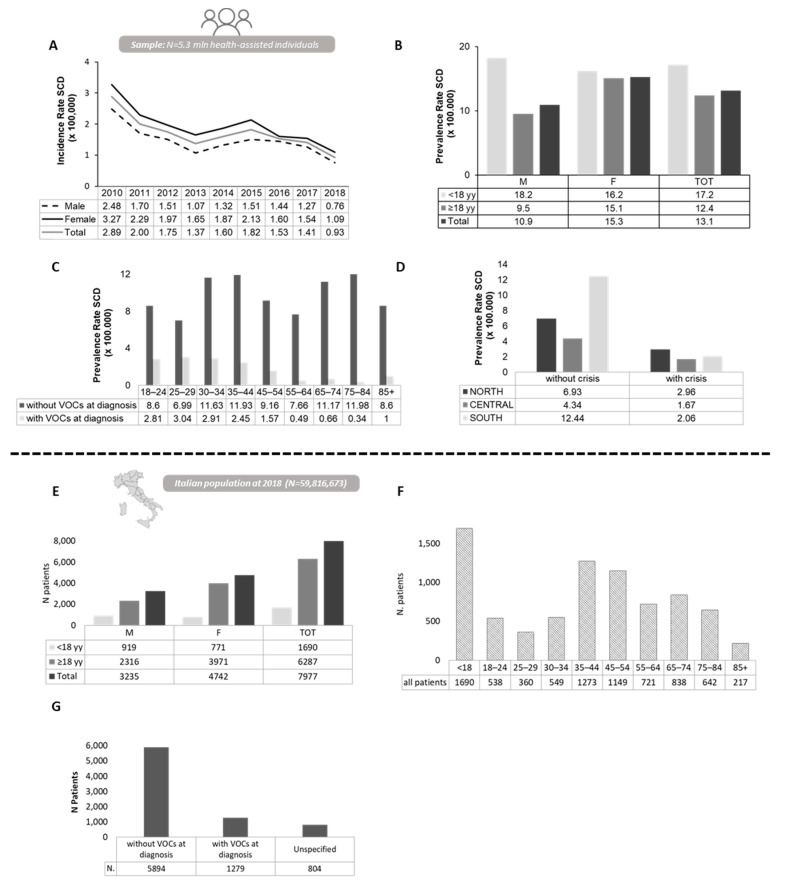
(Upper plots **A**–**D**) Epidemiology of SCD in the sample population (N = 15,300,00). (**A**) Incidence rate overall and by gender per year included in the study (2010–2018). (**B**) Prevalence by gender and age (<18 years and ≥18 years) for year 2018. (**C**) Prevalence by age classes in adult SCD patients with and without VOC at diagnosis for year 2018. (**D**) Prevalence by geographic area and presence of crisis for year 2018. (Lower plots **E**–**G**) Projections to Italian population for year 2018. (**E**) Number of SCD patients estimated in Italy by age and gender. (**F**) Number of SCD patients estimated in Italy by age ranges. (**G**) Number of SCD patients estimated in Italy by presence of VOCs at diagnosis.

**Figure 2 jcm-12-00117-f002:**
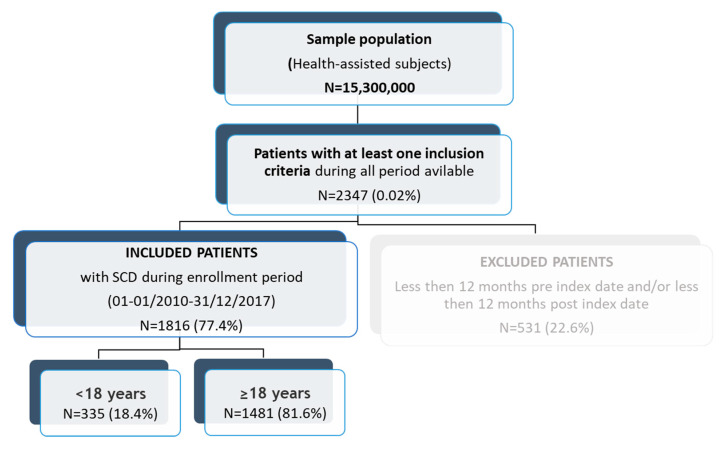
Study flow-chart of patients in analysis. The inclusion period was from 1st January 2010 to 31st December 2017. Patients were identified by SCD diagnosis retrieved from the hospitalization database (ICD-9-CM codes 282.41, 282.42, 282.60, 282.61, 282.62, 282.63, 282.64, 282.68, and 282.69; code 282.50 related to the Sickle cell trait was not considered for inclusion).

**Figure 3 jcm-12-00117-f003:**
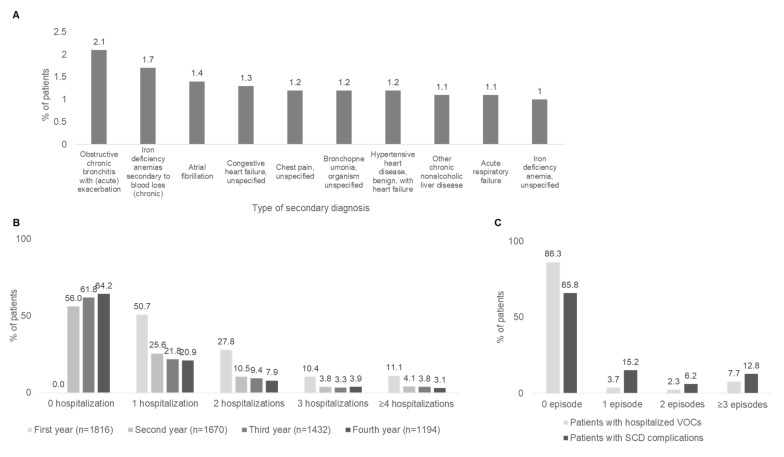
Hospitalization of patients with SCD. (**A**) More frequent primary diagnosis in SCD patients without crisis with secondary SCD diagnosis at inclusion (*n* = 1129). (**B**) Percentages of patients with all-cause hospitalizations during follow-up period. Hospitalizations comprised ordinary admissions and day hospitals; Emergency Room access not requiring hospitalization was not included. Hospitalization corresponding to the index date was counted in the number of hospitalizations. (**C**) Percentages of SCD patients with VOC episodes or complications during follow-up.

**Figure 4 jcm-12-00117-f004:**
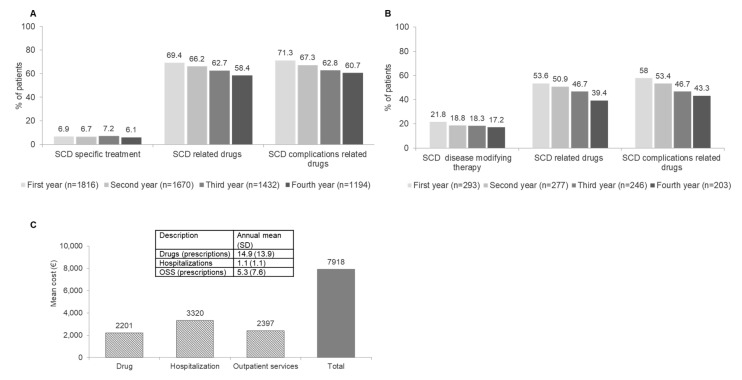
Therapeutic pathways and healthcare resource costs. (**A**) Patients treated with at least one SCD-specific drug, SCD-related drug, and SCD complication-related drug during the follow-up period (1–4 years). (**B**) Patients with crisis treated with at least one SCD-specific drug, SCD-related drug, and SCD complication-related drug during the follow-up period (1–4 years). (**C**) Mean annual healthcare resource consumption and related costs considering all available follow-up. List of SCD treatments in analysis and the related ATC code is reported in Appendix A. Abbreviation: OSS, outpatient services.

**Table 1 jcm-12-00117-t001:** Demographic and clinical characteristics of SCD patients included.

	<18 years (N = 335)	≥18 years (N = 1481)	Total (N = 1816)
Age at index date, mean ± SD	6.9 ± 5.2	52.1 ± 19.9	43.8 ± 25.2
Female, *n* (%)	158 (47.2)	902 (60.9)	1060 (58.4)
Charlson comorbidity index, mean ± SD	0.4 ± 0.5	1.4 ± 1.5	1.2 ± 1.4
Diagnosis with crisis, *n* (%)	88 (26.3)	205 (13.8)	293 (16.1)
Diagnosis without crisis, *n* (%)	181 (54.0)	1168 (78.9)	1349 (74.3)
Diagnosis with unspecified crisis, *n* (%)	66 (19.7)	108 (7.3)	174 (9.6)

**Table 2 jcm-12-00117-t002:** Most frequent ordinary hospitalizations and average length of stays during follow-up period.

MDC	First Year Follow-Up (*n* = 1816)	Second Year Follow-Up (*n* = 1670)	Third Year Follow-Up (*n* = 1432)	Fourth Year Follow-Up (*n* = 1194)
Blood and blood-forming organs and immunological disorders, *n* (%)	290 (16.0)	81 (4.9)	71 (5.0)	45 (3.8)
Average length (in days), mean ± SD	6.9 ± 6.4	5.3 ± 4.3	5.9 ± 5.4	11.6 ± 45.3
Circulatory system, n (%)	218 (12.0)	71 (4.3)	52 (3.6)	31 (2.6)
Average length (in days), mean ± SD	7.9 ± 8.5	8.1 ± 7.2	8.8 ± 6.6	10.4 ± 8.0
Respiratory system, n (%)	197 (10.8)	46 (2.8)	35 (2.4)	41 (3.4)
Average length (in days), mean ± SD	10.2 ± 7.3	10.6 ± 7.4	9.7 ± 5.3	11.1 ± 7.5
Digestive system, n (%)	165 (9.1)	32 (1.9)	27 (1.9)	19 (1.6)
Average length (in days), mean ± SD	7.0 ± 8.8	13.7 ± 19.3	8.0 ± 7.6	6.5 ± 5.2
Hepatobiliary system and pancreas, n (%)	154 (8.5)	36 (2.2)	25 (1.7)	23 (1.9)
Average length (in days), mean ± SD	8.5 ± 7.3	8.3 ± 5.9	8.2 ± 4.6	9.6 ± 6.0
Musculoskeletal system and connective tissue, n (%)	117 (6.4)	42 (2.5)	28 (2.0)	23 (1.9)
Average length (in days), mean ± SD	12.7 ± 13.3	9.1 ± 8.0	8.8 ± 8.8	20.1 ± 56.7
Nervous system, n (%)	91 (5.0)	31 (1.9)	22 (1.5	11 (0.9)
Average length (in days), mean ± SD	13.0 ± 22.9	10.5 ± 11.0	17.6 ± 9.1	13.8 ± 10.8
Kidney and urinary tract *, n (%)	76 (4.2)	25 (1.5)	-	10 (0.8)
Average length (in days), mean ± SD	8.3 ± 6.8	8.9 ± 10.2	-	14.5 ± 11.5
Pregnancy, childbirth, and puerperium **, n (%)	60 (3.3)	-	-	9 (0.8)
Average length (in days), mean ± SD	6.4 ± 8.2	-	-	4.9 ± 3.3
Infectious and parasitic DDs, n (%)	56 (3.1)	20 (1.2)	15 (1.0)	9 (0.8)
Average length (in days), mean ± SD	8.6 ± 12.1	14.3 ± 14.5	9.8 ± 7.5	5.0 ± 3.1

* Not among the top ten more frequent hospitalization during third year of follow-up; ** not among the top ten more frequent hospitalization during second and third year of follow-up. Note: The MDC Blood and Blood-Forming Organs and Immunological Disorders includes SCD diagnosis. List of ICD-9-CM diagnosis codes of sickle cell-related complications is reported in Appendix A.

**Table 3 jcm-12-00117-t003:** Most frequently prescribed drugs during each year of follow-up.

	1st Year Follow-Up (*n* = 1816)		2nd Year Follow-Up (*n* = 1670)		3rd year Follow-Up (*n* = 1432)		4th Year Follow-Up (*n* = 1194)	
Description	*n*	%	*n*	%	*n*	%	*n*	%
Antibacterials for systemic use	1140	62.8	995	59.6	805	56.2	630	52.8
Drugs for acid related disorders	868	47.8	694	41.6	547	38.2	414	34.7
Antithrombotic agents	610	33.6	479	28.7	384	26.8	292	24.5
Anti-inflammatory and antirheumatic products	525	28.9	465	27.8	393	27.4	281	23.5
Antianemic preparations	442	24.3	353	21.1	285	19.9	223	18.7
Drugs for obstructive airway diseases	440	24.2	390	23.4	288	20.1	242	20.3
Agents acting on RAAS	436	24.0	383	22.9	306	21.4	261	21.9
Corticosteroids for systemic use	419	23.1	331	19.8	292	20.4	215	18.0
Diuretics	330	18.2	266	15.9	204	14.2	153	12.8
Beta-blocking agents *	281	15.5	255	15.3	-	-	-	-
Vitamins **	-	-	-	-	241	16.8	200	16.8
Analgesics	209	11.5	188	11.3	134	9.4	104	8.7
Opioids	191	10.5	176	10.0	121	8.4	95	8.0

* Among the top ten more frequent drugs during first and second year of follow-up; ** among the top ten more frequent drugs during third and fourth year of follow-up. Note: List of SCD treatments in analysis and related ATC code is reported in Appendix A.

## Data Availability

All data used for the current study are available upon reasonable request next to CliCon s.r.l. which is the body entitled to allow data treatment and analysis by Local Health Units.

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
