# Peer review of "Real-World Evidence on Disease Burden and Economic Impact of Sickle Cell Disease in Italy"

_jcm, 2022, doi:10.3390/jcm12010117_

Round 1

Reviewer 1 Report

Thank you for your important work. A few, minor points- if it is possible to limit some abbreviations, it might make the methods sections easier to read as reader has to keep going back to abbreviations presented early on. The ICD-9 codes included may be shifted to the supplement rather than main body of text. Some areas of text use casual English and may be made more formal. In addition, are there any quality improvement or cost mitigation studies planned using these data as a platform? This could be helpful to add in the discussion.

Author Response

  • Thank you for your important work. A few, minor points- if it is possible to limit some abbreviations, it might make the methods sections easier to read as reader has to keep going back to abbreviations presented early on.

Reply: we thank the Reviewer for the comment. As suggested, we have removed some acronyms, especially those mentioned only few times. We have maintained anyhow those that are commonly used and well-known in the literature. We have added a section at the end of the manuscript with the list of abbreviation.

  • The ICD-9 codes included may be shifted to the supplement rather than main body of text.

Reply: As suggested, we have moved the ICD-9 codes in the supplementary materials.

  • Some areas of text use casual English and may be made more formal.

Reply: We have addressed this point by an in-depth proofreading.

  • In addition, are there any quality improvement or cost mitigation studies planned using these data as a platform? This could be helpful to add in the discussion.

Reply: We thank the Reviewer for this comment, we have added a sentence in conclusion section.

Reviewer 2 Report

This manuscript aimed to evaluate the prevalence and incidence of sickle cell disease (SCD) in Italy through investigation of the database from the National Health public system and evaluate the economic impact of treatment of these patients. The main contribution was the evaluation of medical costs during treatment and interventions among patients with SCD as the review from Nietert et al., 2002 was focused principally on USA studies and already needs an update. Another strength is the size population analyzed and where it came from, as it is huge and up-to-date.

Concerning aspects related to epidemiological data analyzed, other previous works had incidence and prevalence estimative from many available data such as surveys (Osunkwo et al., 2021), of public databases such as WHO and GHDx (Lippi and Mattiuzzi, 2019), and comparing those previous data numerically with those generated in this research will improve this manuscript. I would like to reinforce that this new analysis is important as it reflects the new world migratory influences from many countries, especially resulting from wars.

In general, this manuscript brings up-to-date data however some points should be clarified to improve the quality. The order in which figures and tables are cited should be adjusted to gain more lecture smoothness. For example, figure supplementary 2 is cited before figure supplementary 1. I also suggest that those figures should be included in the main structure of the paper, with supplementary figure 2s together with figure 1. Supplementary tables 1 and 2 are cited only in the methodology and should be linked to tables 2 and 3, and figure 2 to improve data interpretation and analysis and to link information from those tables that are isolated in this way they are presented. In order to bring more information to the main document and this descriptive data be better appreciated, I suggest creating a new figure aggregating graphics generated from tables 3S, 4S, and 5S to make the manuscript easier to follow without many readings changing to the supplementary and to the main document.

Another question is concerning how Italian population projections from 2018 were made and which statistical methods and programs were employed (figure 2S), as accuracy in the description of how this data was generated will improve data transparency and reproducibility.

A conclusion topic should be explicit and changed from the discussion to a final conclusion topic.

Author Response

  • This manuscript aimed to evaluate the prevalence and incidence of sickle cell disease (SCD) in Italy through investigation of the database from the National Health public system and evaluate the economic impact of treatment of these patients. The main contribution was the evaluation of medical costs during treatment and interventions among patients with SCD as the review from Nietert et al., 2002 was focused principally on USA studies and already needs an update. Another strength is the size population analyzed and where it came from, as it is huge and up-to-date.

Reply: We sincerely thank you for this positive comment.

  • Concerning aspects related to epidemiological data analyzed, other previous works had incidence and prevalence estimative from many available data such as surveys (Osunkwo et al., 2021), of public databases such as WHO and GHDx (Lippi and Mattiuzzi, 2019), and comparing those previous data numerically with those generated in this research will improve this manuscript. I would like to reinforce that this new analysis is important as it reflects the new world migratory influences from many European countries, especially resulting from wars.

Reply: we thank the Reviewer for this comment: in the revised version of the manuscript, we have cited in the discussion the references as suggested.

  • In general, this manuscript brings up-to-date data however some points should be clarified to improve the quality. The order in which figures and tables are cited should be adjusted to gain more lecture smoothness. For example, figure supplementary 2 is cited before figure supplementary 1. I also suggest that those figures should be included in the main structure of the paper, with supplementary figure 2s together with figure 1. Supplementary tables 1 and 2 are cited only in the methodology and should be linked to tables 2 and 3, and figure 2 to improve data interpretation and analysis and to link information from those tables that are isolated in this way they are presented. In order to bring more information to the main document and this descriptive data be better appreciated, I suggest creating a new figure aggregating graphics generated from tables 3S, 4S, and 5S to make the manuscript easier to follow without many readings changing to the supplementary and to the main document.

Reply: As suggested, we have now moved the supplementary figures 1 and 2 in the main text, and incorporated Table 3-5S in a new figure (2A-C). The Figure S3 has been removed since it was redundant, and an explanation embedded in the text (lines 392-395).

  • Another question is concerning how Italian population projections from 2018 were made and which statistical methods and programs were employed (figure 2S), as accuracy in the description of how this data was generated will improve data transparency and reproducibility.

Reply: The projections were made by reproportioning the epidemiological data of the sample population in analysis to national scale, as previously described by our group [Perrone et al. JPPD. 2020;04(03). doi:10.26502/jppd.2572-519X0098; Degli Espost et al, Reumatismo, 2021; 73 (1): 5-14]. We have added this sentence in method section.

  • A conclusion topic should be explicit and changed from the discussion to a final conclusion topic.

Reply: A separate chapter for conclusions was generated as suggested.